# Conformational Defects in the Limbs of Menorca Purebred Horses and Their Relationship to Functionality

**DOI:** 10.3390/ani14071071

**Published:** 2024-03-31

**Authors:** Maria Ripollés-Lobo, Davinia I. Perdomo-González, Mercedes Valera, María D. Gómez

**Affiliations:** 1Departamento de Agronomía, Escuela Técnica Superior de Ingeniería Agronómica, Universidad de Sevilla, Ctra, Utrera Km 1, 41013 Sevilla, Spain; marriplob@alum.us.es (M.R.-L.); dperdomo@us.es (D.I.P.-G.); mvalera@us.es (M.V.); 2Asociación de Criadores y Propietarios de Caballos de Raza Menorquina, Edificio Sa Roqueta C/Bijuters, 36 Bajos, 07760 Ciutadella de Menorca, Spain

**Keywords:** limb defects, equine, genetic parameters, movements, prevalence

## Abstract

**Simple Summary:**

Limb alignments significantly impact the performance and overall well-being of horses, which means that the identification and management of limb-conformation defects is vital for horse owners and breeders. Our study investigates the prevalence of 14 defects in Menorca Purebred horses, along with the environmental factors influencing their occurrence and the genetic parameters for potential inclusion in the official breeding program. Analysis of data from 1120 records from 509 animals reveals a higher prevalence of defects in older females from breeder studs dedicated to breeding, possibly due to limited care. Notably, splay-footed forelimb, closed hocks, camped under, pigeon-toed forelimb and coon foot are the commonest, with a prevalence exceeding that of other equine populations. These defects significantly influence gait scores, particularly in trotting. Heritability estimates range from 0.12 to 0.30, suggesting genetic influence. Genetic correlations show that careful consideration is needed in selective breeding to avoid unintended outcomes. It is therefore advisable to focus selective efforts on the more prevalent defects with higher heritability, as well as to evaluate the inter-defect relationships.

**Abstract:**

Limb-conformation defects significantly influence equine performance and welfare, necessitating thorough investigation for effective management. This study examines the prevalence and genetic parameters of 14 limb-conformation defects in Menorca Purebred horses using data from 1120 records (509 animals with an average age of 101.87 ± 1.74 months) collected between 2015 and 2023. Defects were evaluated using a three-class scale by three appraisers, and a Bayesian approach via Gibbs sampling was employed to estimate genetic parameters including gender, birth period, stud selection criteria, evaluation age and appraiser as fixed effects. *Splay-footed forelimb* and *closed hocks* were the most prevalent defects (67.20% and 62.53%, respectively). Horses with any of the defects analyzed have been observed to obtain significantly lower scores for both walk and trot. Heritability estimates range from 0.12 (s.d.: 0.025) for *closed hock* to 0.30 (s.d.: 0.054) for *base narrow*, confirming the genetic influences on the expression of limb conformation defects. The *divergent defect* in hind limbs showed the highest genetic correlations with forelimb defects (*camped under*, −0.69; s.d: 0.32 and *camped out*, 0.70; s.d: 0.27). The significant genetic correlations between defects highlight the complexity of the relationships, which requires careful consideration.

## 1. Introduction

Selection for suitable morphological qualities for performance in a sport discipline benefits genetic progress by allowing us to make an early pre-selection of the animals, even before they begin their sport career [1]. Deviations from optimal limb structure, commonly known as limb conformation defects or orthopedic diseases, are widespread concerns in equine sporting breeds, and transcend geographical and disciplinary boundaries. 

Furthermore, limb alignments play an important role in sport performance, compromising the horse’s biomechanical efficiency, the horse’s overall well-being and the general health of horse populations. For all these reasons, conformation is a significant indicator of potential soundness and performance, which is pivotal in horse evaluation for purchase, with substantial economic repercussions for owners and breeders, as poor conformation significantly diminishes a horse’s value [2,3,4]. Conformation defects are often caused by underlying complex genetic effects, which may be breed-specific or common in certain types of horses [5].

The economic impact of these defects arises from the diminished athletic capacity and a greater associated risk of musculoskeletal injuries, which can slow down or impede the ideal performance of the animals in different equestrian sports [6]. Changes in conformation shift the center of gravity forward, resulting in variations in gait that impose asymmetrical loads on the musculoskeletal system, predisposing horses to injuries [7]. In this regard, musculoskeletal injury is the most significant cause of wastage in the horse industry [8,9], and accounts for the greater part of culling in riding horses (50–70%) [10,11,12]. Consequently, owners and breeders invest considerable resources in the training and upkeep of their equine athletes, making the identification and mitigation of limb conformation defects crucial for warranting a considerable investment in time and money. Therefore, the primary objective of animal breeders and breeding associations is to improve the genetic potential of the animals over generations [13], with limb conformation serving as a key selection criterion.

Focusing on the common defects shared among different breeds, we aim to explore the potential efficacy of implementing genetic selection as a proactive measure to curb the prevalence of limb-conformation defects, using Menorca Purebred horses (PRMe) as an example. The PRMe is an endangered native breed primarily found in Menorca (Balearic Islands, Spain), and widely known for its role in the regional festivities called “Jaleo Menorquin” [14]. It is a black-coated animal predominantly utilized as a saddle breed, for Classic and Menorcan Dressage. Menorcan Dressage is a different variety of Dressage with specific rules recognized in the Menorca Island. It is a riding modality characterized by its way of holding the reins (using only the left hand), its style of clothing and its horse and rider equipment. The official reprises include typical dressage exercises, but also special Menorca movements (*bot*, in which the horse stands or walks on its hind limbs; *front limb pirouettes*, with the forelimb rotating around the haunches at trot; and *hindlimb pirouettes*, with the hindlimb rotating around the shoulders at trot) [14,15] and they are evaluated at both the recreational and national level [15].

Through a comprehensive examination of limb-conformation defects in the PRMe population, our research seeks to provide valuable insights into the feasibility and impact of genetic interventions for mitigating limb-conformation defects in equine populations dedicated to sport disciplines without losing sight of the main aim of the breeders and breeding organizations, which is to improve the genetic potential of the animals over generations. In this context, our study explores the prevalence of 14 limb-conformation defects that are present in the PRMe population in different degrees, in order to determine the environmental factors that can condition their presence in the population, their influence on walk and trot scores and the genetic parameters (heritability values and genetic correlations) used to evaluate their possible inclusion in the official breeding program. Hopefully, this will provide a promising avenue for breeders and owners to enhance the overall soundness and performance of their equine stock.

## 2. Materials and Methods

### 2.1. Description of Traits and Database

For this analysis, we used data of PRMe horses provided by the Asociación de Criadores y Propietarios de Caballos de Raza Menorquina (ACPCRMe) and collected between 2015 and 2023. A total of 1120 records were recorded from 503 animals (266 males and 237 females, with 605 and 515 records, respectively) belonging to 179 different studs for 14 limb-conformation defects in visits to studs, sport competitions and morphological events featuring PRMe. The average number of records was 2.20 per animal. The animals’ minimum age was 36 months and the maximum was 242 months, with an average age of 101.87 ± 1.74 months. All the individuals included in the analysis were healthy animals, without orthopedic treatments or veterinary interventions.

Limb-conformation defects were evaluated by three different appraisers, previously trained and periodically tested to reduce subjective evaluation, using three classes in a scale, where 0 represents the absence of defect (correct limb conformation), 1 is the slight presence of a defect and 2 the evident presence of a defect. A graphic representation of the defects analyzed and the description of the scale used is shown in Table 1 and the written description is included in Appendix A. Of the animals included, 77.14% were evaluated by more than one appraiser. Additional information of walk and trot scores were also obtained in the corresponding morphological events, by the same appraisers, considering the general characteristics of each gait using a scale of 7 points. Walk score was the mean value for activity, clarity, amplitude and suppleness at walk, while trot score was the mean value for amplitude, suppleness, impulsion, equilibrium and suspension at trot. The walk-score mean value was 5.9 (SD: 1.52) and the trot-score mean value was 5.6 (SD: 1.77).

### 2.2. Statistical and Genetic Analysis

Prevalence of the 14 limb-conformation defects was estimated in a generic way (classes 1 + 2: slight and evident presence of a defect) and for each level (0, 1 and 2: no defect, a slight defect and an evident defect). Prevalence was also estimated by each factor analyzed within a defect: gender (two levels: male and female), birth period (three levels: before year 2000, between 2001 and 2010 and after 2010), stud selection criteria for animals’ purchase or replacement (four levels: (1) functionality (sportive performance), (2) conformation or breed quality, (3) equestrian routes and (4) breeding purpose), evaluation age (two levels: young (<48 months) and adult (≥48 months)) and appraiser (three levels). The stud selection criteria effect was included since the management of the animals in the studs is conditioned by their main use.

A preliminary analysis of variance was carried out to assess the statistical significance of the non-genetic effects that could influence the defect traits analyzed in the PRMe horses. A Generalized Non-Linear Model (GLZ) with a multinomial distribution and a logit link function was used to analyze the effect of gender, birth period, stud selection criteria, evaluation age and appraiser in Statistical for Windows software v.11 [16]. The GLZ represents a methodological approach analogous to a GLM (General Linear Model), with the key distinction being the absence of a continuous Gaussian distribution assumption. Instead, the model assumes other distributions based on the nature of the data being analyzed. Furthermore, to assess the potential impact of limb-conformation defects at different levels on the scores acquired by the animals when both walking and trotting, we employed a GLM followed by a Tukey post hoc test. 

**Table 1 animals-14-01071-t001:** Graphic description of the 14 limb-conformation defects analyzed in the Menorca Purebred horse population, including the definition of the evaluation scale (between 0 to 2).

Defect	Class	Defect	Class
0 (Correct)	2 (Evident)	0 (Correct)	2 (Evident)
Open hockOH	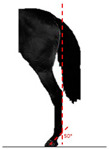	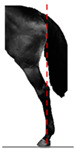	Closed hockCH	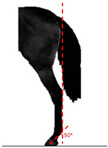	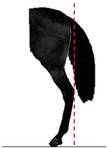
ConvergentConv	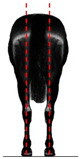	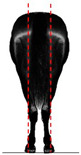	DivergentDiver	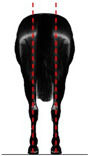	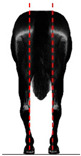
Camped underCU	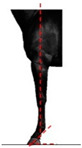	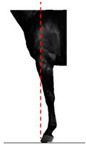	Camped outCO	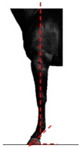	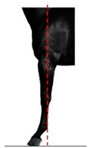
Pigeon-toed forelimbPTF	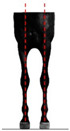	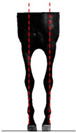	Splay-footed forelimbSFF	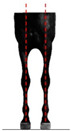	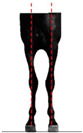
Base narrowBN	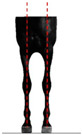	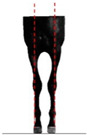	Base wideBW	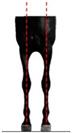	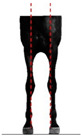
Coon footCF	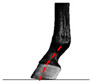	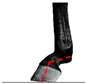	Broken and upright footBUF	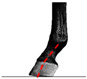	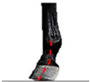
Sloping footSloping	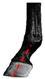	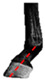	Club footStraight	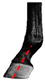	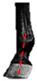

To estimate the genetic parameters (heritability and genetic correlations), we set up a multivariate animal model with a Bayesian approach via Gibbs sampling using GIBBSF90+ and POSTGIBBS modules of the BLUPF90 software [17]. The equation in matrix notation was: **y** = **Xb** + **Zu** + **Wpe** + **e**
where **y** is the vector of observations, **X** the incidence matrix of systematic effect, **Z** the incidence matrix of animal genetic effect, **W** the incidence matrix of random permanent environmental genetic effect, b the vector of systematic effects, u the vector of direct animal genetic effects, pe the vector of random permanent environmental genetic effect and e the vector of residuals. The prior distributions for the systematic effects and the variance components were assumed to be bounded uniform, and the prior distributions for the genetic effects were Gaussian distributions with mean zero and variance defined as follows:(1)$$var\left(\begin{array}{c} u\\ pe\\ e \end{array}\right)=\left(\begin{array}{ccc} A\sigma_{u}^{2} & 0 & 0\\ 0 & A\sigma_{pe}^{2} & 0\\ 0 & 0 & I\sigma_{e}^{2} \end{array}\right)$$
where $\sigma_{u}^{2}$, $\sigma_{pe}^{2}$, and $\sigma_{e}^{2}$ are the additive, random permanent environmental and residual variances. **A** is the numerator relationship matrix and **I** is the identity matrix. More specifically, b included gender (2), birth period (3), stud selection criteria (4), evaluation age (2) and appraiser (3). The Gibbs sampler was run for 250,000 rounds, with the first 50,000 considered as burn-in and then every sample saved for later analysis. Posterior means and standard deviations were calculated to obtain estimates of (co) variance components. Convergence of the posterior distributions generated was assessed using the Geweke’s Z criterion [18], and Monte Carlo sampling error (MCse). The pedigree data for the estimation of the genetic parameters was composed of 1017 animals born between 1961 and 2020 (448 males and 569 females), including between 2 to 5 generations of the animals registered in the official PRMe Studbook, with an average of 2.5 generations.

## 3. Results

The prevalence of the limb-conformation defects analyzed in the PRMe population is outlined in Table 2. Most of the individuals (>50%) are included in the group of unaffected animals (class 0) for most of the defects analyzed, except *closed hock, camped under* and *splay-footed forelimb*, with a lower percentage of animals included in class 0 (37.47%, 44.58% and 32.80%, respectively). 

Notably, the most prevalent defects included *splay-footed forelimb* with an overall prevalence (class 1 and 2) of 67.20%, *closed hock* with 62.53%, *camped under* with 55.42%, *pigeon-toed forelimb* with 44.02%, *coon foot* with 43.81% and *divergent* with 41.90%. The majority of the defects identified are present in a slight form (class 1), with *splay-footed forelimb* and *camped under* being the most frequent defects (58.09% and 50.69% of prevalence for class 1, respectively), and *closed hock* being the limb defect with the highest prevalence in class 2 (evident defect). In contrast, the defects which presented the lowest prevalence values for the evident presence of a defect (all lower than 5%) were *camped out* (0%), *base wide* (1.05%), *sloping* (1.11%), *broken and upright foot* (2.15%), *base narrow and straight* (2.36%), *open hock* (1.46%), *pigeon-toed forelimb* (4.40%) and *camped under* (4.73%).

The relationship of each limb conformation defect with the scores obtained by the animals in the evaluation of walk and trot gaits are shown in the Table 3. Significant differences were detected mainly for the trot scores in the majority of the defects analyzed (64.29%), whereas the walk scores were only influenced by the presence of *splay-footed forelimb*, *coon foot* and *sloping*.

The GLZ (Table 4) revealed that all the effects included in the analysis were significant for some of the limb-conformation defects analyzed, with significance coefficients below 0.05. It is important to emphasize that the appraiser effect was significant for most of the defects included in this analysis (92.86% of the defects analyzed), whereas the evaluation age group was not significant for most of these effects (78.57%), being significant only for *open hock, closed hock* and *convergent*.

The prevalence of the different defects is shown in Appendix A. In general, females presented a higher proportion of limb-conformation defects than males, except for *divergent, splay-footed forelimb, base narrow, coon foot* and *sloping foot*. Also, the older animals (with year of birth ≤ 2000) showed a higher proportion of limb-conformation defects (78.57%) than the other birth-date groups included in the analysis. Surprisingly, animals selected for breeding and conformation (individuals belonging to studs that select their animals for purchase and replacement using breeding and breed-quality criteria) showed the highest prevalence values in most of the traits analyzed (50%), with animals belonging to studs with clear conformation objectives presenting the lowest prevalence values. In line with the results obtained for the year of birth, animals when evaluated as adults presented the highest prevalence values in the majority of the defects analyzed (71.43%). Finally, appraiser number 2 was shown to be the most severe, with the highest prevalence values in 85.71% of the limb defects analyzed, whereas appraiser number 3 showed the lowest prevalence values, and therefore the lowest requirement levels in data collection.

The heritability, estimated variances and convergence parameters values of the defects analyzed are shown in Table 5. The MCse results for all the variables were less than or equal to 0.001, ranging between 0.00053 (*closed hock*) and 0.00116 (*divergent*), and Geweke’s Z-scores for all the variables were included in the range of −2 to +2. All the heritability values were of low-to-medium range, varying between 0.12 (s.d.: 0.025) for *closed hock* and 0.30 (s.d.: 0.054) for *base narrow*. Only 14% of the estimated heritability values were higher than 0.25.

Finally, the genetic correlations between the traits analyzed and their standard deviations are shown in Table 6, of which 64.29% were significant and 40.74% minus values. The 12.96% of the significant correlations can be considered high, with values ≥ 0.50, in absolute value, while 22.22% of them can be considered of a low range, with values ≤ 0.25. The highest correlations were obtained between *camped out* and *divergent* (0.70, s.d.: 0.273), and between *coon foot* and *base narrow* (0.69, s.d.: 0.305) and between *camped under* and *divergent* (−0.69, s.d.: 0.319), the latter being a minus value, while the lowest correlation was obtained between *broken and upright foot* and *splay-footed* forelimb (0.19, s.d.: 0.179).

## 4. Discussion

Limbs are essential in horse functionality and their conformation affects the horse’s susceptibility to injury [2], especially to musculoskeletal injuries, which are the most significant cause of financial loss in the horse industry [8,9]. At the same time, limb conformation is a key factor used by buyers when evaluating horses, where straight conformation is preferred [19]. Thus, the identification of potential predisposing factors, such as specific conformational abnormalities, is a worthwhile aim, given considerable savings in time and money it can bring [2].

Morphological defects in any species, including horses, have origins that are both polygenic [20,21] and multifactorial [22], and selection is usually based on data collected during the evaluation of an animal’s conformation by technicians and/or appraisers under field conditions. Angular limb deformities usually are of congenital origin, but they can also be developmental and have numerous causes, such as genetics, nutrition, amount of exercise, weight bearing or veterinary interventions, among others, which can all influence the conformation of the adult horse [23].

Here, we analyzed limb-conformation defects in the PRMe population in order to evidence the prevalence of these problems and evaluate their possible inclusion in the official breeding program of this population. Thus, according to the results obtained (Table 2), the most common defects were found to be *splay-footed forelimb*, *closed hocks*, *camped under*, *pigeon-toed forelimb* and *coon foot,* and, on the whole, the prevalence percentages obtained in the population analyzed are higher than those reported in other equine populations for the same defects. As has been observed in Pura Raza Española horses [24], with a prevalence of 74.12%, *convergent* hock defect is one of the most prevalent limb defects in the population analyzed in our study, albeit with a lower prevalence (38.83%). *Closed hock* is also a common defect in the PRMe population, showing lower prevalence values than the Pura Raza Española horses analyzed [5] (22.09%). Furthermore, lower prevalence rates were reported for *closed hock* in Swedish Warmbloods (4.30%; [25]) and Thoroughbred horses (31%; [26]). The Thoroughbred breed exhibited a prevalence of 30.10% for *splay-footed forelimb* defect and 19.39% for *pigeon-toed forelimb* defect [2], both of which are lower than those obtained in the current analysis. Additionally, *camped under* shown a higher prevalence in the PRMe population than in Swedish Warmbloods (5.60%; [25]), but lower than in Thoroughbred horses (58%; [26]). Various methodologies are utilized to investigate conformational defects in horses, encompassing visual evaluation, physical examination, static conformation analysis, dynamic assessment of movement, gait analysis, diagnostic imaging, and computer-aided modeling. It is worth mentioning that, in the above work mentioned, while Swedish Warmbloods horses were evaluated directly on the animals [25], the data obtained from Thoroughbred horses [26] were made by photographic images that were later analyzed. On the other hand, in Pura Raza Española horses [5], these defects were analyzed in a similar way to the PRMe, with a three-level scale, where 0 is the absence of defect and the other levels reflect the presence of defect in different levels (1 is scarce and 2 is evident). These approaches collectively offer insights into skeletal alignment, musculoskeletal health and biomechanical performance, crucial for assessing the impact of conformational abnormalities on the horses’ soundness and athletic potential. 

The relationship between walk and trot and conformational defects in sport horses is important for understanding their performance capabilities. Conformational abnormalities can significantly impact movement biomechanics, leading to irregularities in gaits such as the walk and trot [5]. Horses with structural deviations may experience choppy or uneven gaits. Furthermore, certain conformational defects can predispose horses to lameness issues or musculoskeletal injuries, particularly during high-intensity activities like trotting or galloping, thus affecting their soundness and long-term athletic potential. A poor stance of the forelimbs predisposes horses to lower soundness and durability, due to the role of heels and pasterns as shock absorbers during movement [27]. Toe-in and toe-out appearance, which often accompanies angular deformities, is a concurrent rotational deformity [23]. In this context, the prevalence of *splay-footed forelimb* and *pigeon-toed forelimb* in PRMe is more than double that of the Thoroughbreds analyzed [2]. These authors highlighted the relationship of both limb defects with poor racing performance in horses with severe defects. In this regard, the limb-conformation defects analyzed in the PRMe population exert a greater influence on trotting than walk scores (Table 3). The presence of *splay-footed forelimb* affects both walk and trot scores, while *camped under, base narrow* and *base wide* are the only defects without any influence on either of these gait scores. We can therefore conclude that limb-conformation defects significantly impact gait scores in PRMe individuals, as they are crucial factors in their use for riding, in which dressage ability is conditioned by the quality of the gait.

The significant effect of forelimb toe-in and toe-out on the likelihood of contracting carpitis has been demonstrated [28]. Additionally, toe-in and toe-out had the strongest phenotypic effects on health from an orthopedic and locomotive standpoint, along with straight pasterns and changes in lateral hock angles [29]. Furthermore, stride length, smoothness of gait, soundness of limbs and power of propulsion depend on the structure of the forelimbs [30,31], with alterations in the foot axis (pastern-hoof) increasing cases of concussion, stilting the horse’s action and diminishing its spring and freedom of gait, which are crucial features of stride [31,32]. Therefore, the significant prevalence detected for *coon foot* is a key finding in the analysis of this population, due to its negative effects on performance, health and soundness.

The hock plays an important role in propulsion and helps to decrease the harmful effects of concussion, as it is the region where the propulsive efforts of the extensor muscles are concentrated [32]. The negative effects of *convergent* hock have been described [31], causing joint wear from fatigue, reduced stride and greater stress on the plantar ligaments at the rear of the hock. This defect can lead to a narrow rump, accompanied by weak muscles, and may result in horses having little resistance to constant, high-intensity exercises [33] and experiencing greater difficulty in achieving correct movements [34]. In extreme cases, the hocks may rub against each other, thus hindering movement [5] and leading to stress on the outside of the hocks [31]. The higher tendency of PRMe horses towards *convergent* hocks may be related with prolonged training for the *bot*. In this exercise, the horse raises its forelimbs and stands or walks on its hind limbs [14] to perform the traditional Menorcan movements in Menorcan Dressage, an exercise which is awarded with a high score when it is performed vertically and is well-balanced, with the animals walking or jumping in this position.

The consequences of *camped under* on movement in Pura Raza Español stallions have been analyzed [24], and the animals with this limb defect experience a delayed takeoff, prolonged stance periods, reduced swing phases and decreased stride length, predisposing them to stumbling and suffering injuries due to increased pressure in the navicular zone, which may, in turn, lead to bone damage and weakening of the tendons and ligaments [33]. Previous analysis [28] also demonstrated the significant effect of *camped under* on the likelihood of suffering carpitis, while other authors [26] showed that horses with the *camped under* defect exhibit an increased elbow-joint lateral angle and decreased fore-fetlock-joint lateral angle.

Pastern angle is very important in determining the amount of load on the lower-limb structures [35], as the forelimb carries most of a horse’s weight (60–65%) [31]. Here, some authors [28] confirmed an increased incidence of overall and bilateral carpitis in animals with anterior broken toe-axis. To assess the impact of limb-conformation defects, it is crucial to take into account the fact that racehorses and riding horses engage in different activities during their sport uses. Therefore, the significance of the different conformational abnormalities analyzed may vary across breeds, depending on their sport or recreational use [29].

As Table 4 reveals, all the effects analyzed are significant for some of the limb conformation defects included in the analysis (*p* < 0.05). For instance, the appraiser effect is significant for 92.86% of the defects analyzed, whereas age group is significant for only 21.43% of them. Depending on the appraiser, significant changes in prevalence can be seen, with appraiser 2 being the strictest in determining the existence of limb-conformation defects, (Appendix A). The appraiser/judge effect has previously been evidenced in the evaluation of conformation traits in other horse populations, such as the Pura Raza Española, using the whole range of scores and collecting the maximum level of variation both in the application of linear assessment [36] and in the official conformation contests [37]. 

A higher proportion of limb-conformation defects is observed in females born in the year 2000 or before and belonging to breeder studs dedicated to breeding and conformation. These results can be explained by lower selection pressure and fewer management interventions, such as periodical horseshoeing, applied to a greater number of dams compared to sires in the PRMe population, which may influence the visual evaluation of limb-conformation defects by appraisers. Furthermore, an improvement in prevalence data is evident in the younger generation, with a noticeable decrease in prevalence when comparing animals born in different birth periods included in the analysis, with the exception of *splay-footed forelimb*, which shows no improvement in prevalence data between the periods analyzed and remains a constant defect observed in the study population.

Regarding the selection criteria in studs, those that select their animals based on conformation criteria, such as in conformation contests that evaluate the breed quality of the animals compared to the breed standard, showed lower prevalence values for 50% of the defects analyzed. This is because is their primary selection criterion is conformation excellence, which influences the management and constant care given to their animals. In contrast, studs that select animals only for breeding purposes show higher prevalence values for 50% of the defects analyzed, mainly due to the fewer management interventions in mares in the breeding system in this population: they are managed in an extensive production system, with limited management, little professional care and few supplementary controls, except when necessary for health, welfare and reproductive interventions (such as medication to control parasites, estrus and pregnancy veterinary controls, identification of foals, or supplementary feeding as required, among others). In general, reproductive mares go barefoot, and their hooves receive limited care and poor handling, which can affect the visual evaluation of limb-conformation defects by appraisers.

Due to the negative effects of these limb defects in performance, health and soundness, selective action is recommended to improve limb conformation in future generations. The estimation of genetic parameters reveals medium-to-low heritability values for the defects analyzed (see Table 5), demonstrating the genetic determination of these defects. In general, low heritability values have been reported by most of the authors reviewed, except in Pura Raza Española horses (between 0.14 and 0.42) [5] and in Belgian Warmblood horses (ranging between 0.15 and 0.55) [38]. Low heritability values for global forelimb (0.12) and hindlimb (0.15) conformation have also been reported [39].

When the most prevalent defects in the PRMe population are analyzed, *closed hock* and *camped under* defects show heritability values that are lower than those reported in the reviewed bibliography for Pura Raza Española horses (0.40 [5]) and Belgian Warmblood horses (0.35 [38]), respectively; and *pigeon-toed* and *splay-footed forelimbs* defects show higher heritability values in comparison to the Thoroughbreds analyzed [2] (0.17 and 0.16, respectively), the latter authors also reporting higher heritability values for *open hock* (0.38) and *sloping* (0.31), lower values for *base narrow* (0.16), and the same heritability value for the *straight* defect (0.18) in Thoroughbred horses.

Different authors [38,40] evaluated the defects using a linear scale that includes the opposite defects in the extremes of the same scale, and lower heritability values were recorded in the five limb-conformation defects included in their study, which ranged between 0.04 for the hind pastern and 0.14 for the formation of hind legs [40]. However, this leads to an error in estimation of heritability values, as defects should be assessed independently rather than in combination, as occurs with the use of the linear scale, because the genes responsible for one defect may not coincide with the major genes acting on an opposite defect, as suggested other authors [5]. These authors reported heritability values of 0.14 and 0.22, respectively, for opposite defects such as *open* and *closed hock*, similar to those values obtained in our study for the same defect separately (0.16 and 0.12, respectively). Also, in the analysis of the opposite defects *sloping* and *straight*, heritability values of 0.09 and 0.25 were found, respectively, whereas the PRMe horses showed heritability values of 0.11 and 0.18 for each trait. Analysis of the Belgian Warmblood horse [38] also assessed *convergent-divergent* defects (0.24) and *camped under-camped out* defects (0.35), obtaining heritability values in range with those obtained in the present study for *convergent* (0.24) and *divergent* (0.18), and higher than those obtained in the PRMe population for *camped under* (0.17) and *camped out* (0.25). Additionally, the Warmblood Sport horses in the Czech Republic analyzed [40] have shown the opposite defects, *coon foot* and *broken and upright foot*, obtaining a heritability value of 0.13, lower than those obtained in the population analyzed when each defect is analyzed separately (0.20 and 0.23, respectively).

Finally, the majority of genetic correlations between the traits analyzed (Table 6) were significant (64.29%), ranging between 0.20 and 0.70, a similar range to that reported in Pura Raza Español horses (0.13–0.70 [5]) and slightly higher than that reported for Warmblood Sport horses in limb-conformation traits (0.02–0.62 [40]). Notably, 40.74% of the genetic correlations obtained were minus values, which must be taken into account when implementing selective action in the breeding program of this population, as efforts to reduce the presence of one defect may inadvertently increase the presence of others. The *divergent* defect in hind limbs showed a high genetic correlation with the forelimb defects *camped under* (−0.69; s.d.: 0.319) and *camped out* (0.70; s.d.: 0.273), which are opposite defects. These values indicate that the presence of *divergent* defects in the hind limbs is associated with a lesser presence of *camped under* and a greater presence of *camped out* in the forelimbs. A lower value was shown for the correlation between *divergent* and *camped under* defects in Pura Raza Español horses (0.23 [5]). In addition, the correlation between *base wide* and *coon foot* in forelimbs was also high and positive (0.69; s.d.: 0.305), suggesting that selection efforts to reduce the presence of one of these may also contribute to a reduction in the other. The defects with the highest prevalence values for the ‘evident defect’ class (class 2) in the PRMe population showed genetic correlations of medium range with the other defects analyzed. The highest correlation was obtained between *splay-footed forelimb* and *straight* (0.50; s.d.: 0.436), while the lowest correlation was between *splay-footed forelimb* and *broken and upright foot* (0.19; s.d.: 0.179). Therefore, selection efforts targeting certain limb-conformation defects can contribute to a lesser extent to the improvement of others. 

Finally, despite the limitations encountered in this analysis (such as a low number of sampled animals, traits evaluated subjectively and a limited number of known generations in the pedigree), the heritability of limb-conformation defects and their genetic correlations suggest promising opportunities for defect improvement and, to some extent, health enhancement through selective breeding. However, only limited improvements in the prevalence of certain limb defects may be expected, particularly when heritability and prevalence values are low. It is important to note that the Menorca horses are recognized as an endangered population, and their official breeding program includes measures for their conservation and selection [41]. Within this context, each mating must be authorized through a certificate that guarantees that a given individual does not exceed the maximum number of offspring to preserve the genetic variability of the population. To achieve this goal, the genetic variability of the Menorca horse population is monitored annually, using pedigree and molecular data, to guide breeders in the genetic management of the population. In other words, the Menorca Purebred horse breeding program serves a dual purpose with a low intensity of selection.

## 5. Conclusions

*Splay-footed forelimb, closed hock*, *camped under*, *pigeon-toed forelimb*, *coon foot* and *divergent* are the most prevalent limb-conformation defects in the PRMe population, all of which are influenced by different external factors, with the stud selection criteria being one of the most important. The effect of limb-conformation defects on sport use in the animals has been evidenced by the scores obtained at walk and trot. To reduce the incidence of these defects in this population used for riding and dressage, selective efforts must be focused against the most prevalent defects. Nevertheless, to include various defects effectively, it is vital to evaluate the relationships between them to avoid the efforts proving redundant or counter-productive in the official breeding program.

## Figures and Tables

**Table 2 animals-14-01071-t002:** Prevalence of the 14 limb-conformation defects analyzed in Menorca Purebred horses according to the class.

Limb Defect		N	Class							
			0		1		2		1 + 2	
			n	%	n	%	n	%	n	%
Open hock	OH	343	238	69.39	100	29.15	5	1.46	105	30.61
Closed hock	CH	483	181	37.47	233	48.24	69	14.29	302	62.53
Convergent	Conv	412	252	61.17	122	29.61	38	9.22	160	38.83
Divergent	Diver	432	251	58.10	148	34.26	33	7.64	181	41.90
Camped under	CU	507	226	44.58	257	50.69	24	4.73	281	55.42
Camped out	CO	356	341	95.79	15	4.21	0	0	15	4.21
Pigeon-toed forelimb	PTF	318	178	55.97	126	39.62	14	4.40	140	44.02
Splay-footed forelimb	SFF	439	144	32.80	255	58.09	40	9.11	295	67.20
Base narrow	BN	467	374	80.09	82	17.56	11	2.36	93	19.92
Base wide	BW	476	363	76.26	108	22.69	5	1.05	113	23.74
Broken and upright foot	BUF	418	311	74.4	98	23.44	9	2.15	107	25.59
Coon foot	CF	477	268	56.18	184	38.57	25	5.24	209	43.81
Club foot	Straight	467	313	67.02	143	30.62	11	2.36	154	32.98
Sloping foot	Sloping	452	310	68.58	137	30.31	5	1.11	142	31.42

Abbreviations of limb-conformation defects are listed in Table 1. Key: the number of animals is indicated as N or n, class 0 is no defect, class 1 is slight defect and class 2 is evident defect.

**Table 3 animals-14-01071-t003:** Generalized Linear Model and post hoc analysis (Tukey LSD) of the scores obtained by the animals for walk and trot in the official evaluations within the breeding program, depending on the level of each of the defects analyzed in the Menorca Purebred horses.

Defect	Level	N	Walk	Trot	Defect	Level	N	Walk	Trot
OH	1	231	5.99	5.97 ^b^	SFF	1	143	6.07 ^b^	5.75 ^b^
	2	98	5.74	5.37 ^a^		2	252	5.89 ^b^	5.56 ^ab^
	3	5	6.84	5.29 ^ab^		3	40	5.18 ^a^	4.94 ^a^
CH	1	175	5.93	5.84 ^b^	BN	1	369	5.89	5.61
	2	231	5.95	5.51 ^b^		2	83	5.51	5.21
	3	67	5.48	4.85 ^a^		3	10	5.52	5.74
Conv	1	244	5.95	5.74 ^b^	BW	1	358	5.88	5.62
	2	122	5.77	5.43 ^ab^		2	108	5.86	5.30
	3	37	5.37	4.89 ^a^		3	5	6.63	5.29
Diver	1	244	5.86	5.53 ^ab^	BUF	1	308	5.98	5.77 ^b^
	2	147	5.97	5.79 ^b^		2	98	5.78	4.98 ^a^
	3	33	5.91	4.97 ^a^		3	9	5.84	4.72 ^ab^
CO	1	337	5.96	5.70 ^b^	CF	1	263	6.02 ^b^	5.64
	2	15	5.61	4.83 ^a^		2	183	5.64 ^a^	5.46
	3	0	-	-		3	25	5.60 ^ab^	5.62
CU	1	221	5.85	5.70	Sloping	1	305	5.98 ^b^	5.70
	2	256	5.88	5.42		2	136	5.56 ^a^	5.29
	3	24	5.65	5.01		3	6	5.96 ^ab^	5.61
PTF	1	175	6.12	6.06 ^b^	Straight	1	308	5.91	5.73 ^b^
	2	126	5.88	5.22 ^a^		2	143	5.78	5.17 ^a^
	3	14	5.56	5.42 ^ab^		3	11	5.07	4.39 ^a^

Abbreviations of limb-conformation defects are listed in Table 1. Different superscript letters (a and b) indicate a statistically significant difference between groups (*p* < 0.05).

**Table 4 animals-14-01071-t004:** Generalized Non-linear Model for the 14 limb-conformation defects analyzed in the Menorca Purebred horses based on their non-genetic effects (*p*-value).

Defects	Effects				
	Gender	Birth Period	Stud Selection Criteria	Evaluation Age	Appraiser
OH	0.965	<0.001	0.388	0.009	0.068
CH	0.699	0.031	0.005	0.015	<0.001
Conv	0.689	0.005	<0.001	0.004	<0.001
Diver	0.071	<0.001	0.355	0.387	0.001
CU	0.115	0.005	0.015	0.414	<0.001
CO	0.615	0.047	0.410	0.333	<0.001
PTF	0.283	0.123	<0.001	0.134	0.002
SFF	<0.001	0.315	<0.001	0.368	<0.001
BN	0.079	0.323	0.027	0.347	<0.001
BW	0.213	0.018	0.125	0.344	<0.001
BUF	0.003	<0.001	0.499	0.387	<0.001
CF	<0.001	0.005	0.077	0.823	0.001
Straight	0.415	<0.001	0.271	0.431	<0.001
Sloping	0.010	<0.001	0.021	0.449	0.034

Abbreviations of limb-conformation defects are listed in Table 1.

**Table 5 animals-14-01071-t005:** Genetic parameters for limb-conformation defects analyzed in the Menorca Purebred horse population.

Defects	σ_pe_			σ_u_			σ_e_			h^2^ (s.d.)	HPD 95%
	Mean	Median	HPD 95%	Mean	Median	HPD 95%	Mean	Median	HPD 95%		
OH	542.23	539.20	373.60–715.00	367.04	363.50	224.10–515.30	1415.60	1413.00	1264.00–1573.00	0.16 (0.029)	0.100–0.215
CH	276.32	275.00	195.90–359.90	138.89	137.20	77.410–201.80	784.17	782.70	705.10–870.80	0.12 (0.025)	0.067–0.165
Conv	631.57	630.60	444.80–814.90	487.25	478.00	282.80–709.90	944.30	942.30	839.80–1050.00	0.24 (0.047)	0.147–0.329
Diver	593.19	592.00	406.30–788.40	308.43	300.50	142.40–488.20	853.16	851.30	757.70–951.90	0.18 (0.048)	0.086–0.270
CU	26.07	25.83	15.43–37.12	32.03	31.52	17.60–48.11	132.83	132.60	119.30–147.20	0.17 (0.037)	0.095–0.240
CO	461.32	457.00	286.60–638.30	584.53	576.20	349.40–831.00	1309.90	1307.00	1166.00–1457.00	0.25 (0.045)	0.161–0.335
PTF	903.71	900.20	662.20–1146.00	618.70	609.40	360.10–888.30	1046.00	1044.00	929.60–1163.00	0.24 (0.046)	0.152–0.332
SFF	636.58	634.20	452.30–824.60	374.52	365.80	192.80–570.30	755.11	753.50	673.00–840.90	0.21 (0.051)	0.117–0.311
BN	267.23	265.80	170.80–362.70	384.72	379.60	230.60–553.60	633.99	632.70	564.20–705.80	0.30 (0.054)	0.196–0.401
BW	270.58	269.00	177.80–367.40	315.13	310.90	189.10–447.90	584.43	583.30	521.70–649.40	0.27 (0.048)	0.175–0.363
BUF	457.33	454.50	294.50–619.00	477.58	472.10	281.20–681.20	1124.10	1122.00	1008.00–1251.00	0.23 (0.044)	0.147–0.316
CF	294.29	292.10	195.80–397.90	240.99	236.90	137.40–352.40	692.73	691.60	622.10–767.90	0.20 (0.040)	0.118–0.274
Straight	342.65	339.20	222.10–469.90	288.71	283.90	162.20–422.90	940.42	938.70	840.90–1039.00	0.18 (0.038)	0.110–0.258
Sloping	411.64	408.70	283.30–547.30	368.72	362.90	230.00–517.80	906.70	905.00	813.20–1006.00	0.22 (0.038)	0.145–0.293

Abbreviations of limb-conformation defects are listed in Table 1. σ_pe_: additive genetic variances of random permanent environmental effect; σ_u_: additive genetic variances of animal; σ_e_: residual variances; HPD 95%: 95% higher posterior density; h2: heritability value; s.d.: standard deviation.

**Table 6 animals-14-01071-t006:** Genetic correlations and standard deviation (between brackets) among the 14 limb-conformation defects analyzed in Menorca Purebred horses.

	Conv	Diver	CU	CO	PTF	SFF	BN	BW	BUF	CF	Straight	Sloping
OH	−0.36 (0.127)	0.54 (0.127)	0.38 (0.153)	−0.20 (0.143)	0.13 (0.152)	−0.23 (0.154)	0.49 (0.134)	0.31 (0.464)	−0.14 (0.160)	0.40 (0.158)	0.36 (0.466)	0.55 (0.128)
CH	0.23 (0.145)	−0.45 (0.149)	−0.21 (0.434)	0.07 (0.152)	0.28 (0.154)	0.12 (0.247)	−0.49 (0.143)	0.09 (0.170)	0.48 (0.140)	−0.35 (0.164)	0.26 (0.167)	−0.07 (0.106)
Conv	-	-	0.14 (0.366)	−0.30 (0.153)	−0.26 (0.158)	−0.26 (0.368)	−0.33 (0.099)	0.24 (0.183)	−0.11 (0.190)	−0.39 (0.296)	−0.31 (0.184)	−0.12 (0.174)
Diver	-	-	−0.69 (0.319)	0.70 (0.273)	−0.33 (0.346)	0.37 (0.326)	0.28 (0.192)	−0.19 (0.192)	−0.22 (0.190)	0.28 (0.200)	−0.33 (0.401)	0.43 (0.154)
CU			-	-	−0.20 (0.158)	0.21 (0.184)	0.20 (0.171)	0.02 (0.188)	0.42 (0.380)	0.05 (0.187)	−0.26 (0.168)	0.24 (0.178)
CO			-	-	0.21 (0.152)	−0.09 (0.171)	−0.10 (0.427)	−0.33 (0.150)	0.09 (0.162)	0.08 (0.169)	0.11 (0.173)	−0.06 (0.389)
PTF					-	-	0.32 (0.139)	−0.39 (0.146)	−0.35 (0.320)	0.08 (0.169)	−0.28 (0.465)	0.33 (0.300)
SFF					-	-	−0.24 (0.161)	0.36 (0.163)	0.19 (0.179)	0.42 (0.440)	0.50 (0.436)	−0.26 (0.474)
BN							-	-	−0.21 (0.154)	0.42 (0.440)	0.21 (0.310)	0.65 (0.300)
BW							-	-	0.27 (0.155)	0.69 (0.305)	0.44 (0.380)	0.26 (0.146)
BUF									-	-	−0.47 (0.133)	0.46 (0.146)
CF									-	-	−0.23 (0.161)	0.48 (0.123)

Abbreviations of limb-conformation defects are listed in Table 1.

## Data Availability

The data base with conformational defects, the environmental effects included in the models and the pedigree information are available online: https://zenodo.org/records/10705461 (accessed on date 26 February 2024).

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
