# Peer review of "Conformational Defects in the Limbs of Menorca Purebred Horses and Their Relationship to Functionality"

_animals, 2024, doi:10.3390/ani14071071_

Round 1
Reviewer 1 Report
Comments and Suggestions for Authors
In their morphological study, Ripollés-Lobo and colleagues statistically evaluated the prevalence of limb conformation defects in Menorca Purebred Horses. It is an interesting look at the problem of the use of these animals, but exclusive on the part of owners and breeders. The work is of limited scientific value (without an experimental arrangement), and consider it more of a descriptive work.
Specific comments.
1. Please provide an affiliation in English.
2. Line 100 - The description of the health status of the group of animals studied is not very precise. First, the age of the oldest of the animals is missing. In addition, there is no information on whether the animals have undergone orthopedic treatment. Whether any veterinary interventions (fractures, dislocations) in the limbs took place.
3. Line 102 - How did the appraisers average their assessments of changes in limb structure?
4. Line 117 - It is unclear whether the authors checked the normal distribution of the data before performing the analysis of variance test.
5. Please indicate exactly what genetic parameters were evaluated?
6. Please check the correctness and validity of the use of abbreviations. For example, the abbreviation GLM (line 124) is used only once in the main text. The abbreviation GLZ is defined as many as twice (line 119 and 180).
7. Please avoid using names in references. The journal's recommendation is to use numbers only.
Author Response
REVIEWER 1
In their morphological study, Ripollés-Lobo and colleagues statistically evaluated the prevalence of limb conformation defects in Menorca Purebred Horses. It is an interesting look at the problem of the use of these animals, but exclusive on the part of owners and breeders. The work is of limited scientific value (without an experimental arrangement), and consider it more of a descriptive work.
Answer: Firstly, the authors express gratitude for your assistance in enhancing this manuscript. All revisions, guided by the reviewers' recommendations, have been incorporated into the revised version, highlighted in red color.
The study commences with an assessment of the prevalence of limb conformation defects in Menorca Purebred horses, aiming to ascertain the relevance and necessity of this investigation within the reference population. Subsequently, after evaluating potential areas for improvement and identified needs, a comprehensive genetic analysis of limb conformation traits is conducted. This analysis includes estimating heritability values and assessing genetic correlations among the traits, with the aim of determining their potential integration into the official breeding program for the Menorca population. Finally, the conclusion section offers pertinent recommendations in this sense.
It is noteworthy that there is a dearth of scientific publications on conformation defects in horses, with the exception of the study by Ripollés-Lobo et al. (2023), which addresses different defects. Consequently, we consider that our analysis could serve as an international reference for other populations. These reasons are remarked in the introduction section (lines 80-91).
Specific comments.
- Please provide an affiliation in English.
Answer: We have reviewed the guidelines for authors and found no specific stipulations regarding language usage in this section. Additionally, upon examining other publications in Animals, we observed that affiliations are presented in Spanish without problem. Hence, we opt to retain the Spanish language for affiliations.
- Line 100 - The description of the health status of the group of animals studied is not very precise. First, the age of the oldest of the animals is missing. In addition, there is no information on whether the animals have undergone orthopedic treatment. Whether any veterinary interventions (fractures, dislocations) in the limbs took place.
Answer: Required information is included in the new version (lines 104-107).
- Line 102 - How did the appraisers average their assessments of changes in limb structure?
Answer: Appraisers have been trained before data collection and their evaluations are evaluated periodically to avoid mistakes or inconsistencies. This information is included in the new version (lines 108-109).
- Line 117 - It is unclear whether the authors checked the normal distribution of the data before performing the analysis of variance test.
Answer: Of course, the normality of the dependent variables was verified. Since none of them followed the normal distribution, we decided to analyze the statistical significance of the possible factors to be included in the genetic parameters’ estimation model using a GLZ model (Generalized Linear/Nonlinear Models). GLZ is a generalization of the General Linear Model (GLM) which is able to address both linear and nonlinear effects for any number and type of predictor variables on a discrete or continuous dependent variable. Therefore, the GLZ represents a methodological approach analogous to a GLM, with the key distinction being the absence of a continuous Gaussian distribution assumption. In our study, the GLZ has been fitted to a Multinomial distribution (solved by a logit link function). In the same way, to obtain the genetic parameters, we use a Bayesian approach, which neither assumes a specific type of distribution for the analyzed traits. This information was included in the methodology section of the manuscript (lines 131-141).
- Please indicate exactly what genetic parameters were evaluated?
Answer: Required information is included in the new version (line 142)
- Please check the correctness and validity of the use of abbreviations. For example, the abbreviation GLM (line 124) is used only once in the main text. The abbreviation GLZ is defined as many as twice (line 119 and 180).
Answer: This paragraph has been re-written and improved.
- Please avoid using names in references. The journal's recommendation is to use numbers only.
Answer: This suggestion has been reviewed in the new version, following journals’ recommendations.
Reviewer 2 Report
Comments and Suggestions for Authors
The overall qaulity of the manuscript is fine. After reading of the manuscript, I have only a few suggestions to make minor changes/clarifications.
- Simple Summary is not in agreement with the main text as in L19 the heritability intervals are differs from Abstract and the data reported in Table 5.
- The signs of significant differences in Table 3 must be uniform. Most of the letterings start from highest to lowest, so it should be corrected in this way.
Author Response
REVIEWER 2
The overall quality of the manuscript is fine. After reading of the manuscript, I have only a few suggestions to make minor changes/clarifications.
Answer: Authors wish to thank you for your help to improve this manuscript. All the changes realized following the recommendations of the reviewed are included in the new version using red color.
- Simple Summary is not in agreement with the main text as in L19 the heritability intervals are differs from Abstract and the data reported in Table 5.
Answer: Thank you. It was a mistake. Required information has been changed in the new version (line 19).
- The signs of significant differences in Table 3 must be uniform. Most of the letterings start from highest to lowest, so it should be corrected in this way.
Answer: This information has been changed in the new version (table 3).
Reviewer 3 Report
Comments and Suggestions for Authors
Animals – 2924794
The paper covers a very interesting subject on specific horse evaluation. The paper is well written and well structured. However, some important items have to be corrected and explained. The main issue is connected with traits presentation and lack of the limitation part in the discussion. Table 1/ (figure rather) with graphical traits description traits seems not clear in all cases, evaluation and conditions of walk and trot, being also evaluated traits, are not described. Some sentences need explanation.
In detail:
L 72-75 Please specify the movements and the practical source of their description, these terminology – bot, spine do not come from FEI dressage rules – and you call your horse - dressage horse.
This terminology –spine - can be seen as Western riding connected – as spine is a Western term, and the readers should know what you mean exactly. Your explanations even interesting, are too short and not enough. Be more clear in the paper and give citations to the Menorca horse dressage rules/or detailed descriptions of movements you describe.
Table 1 – it should be clearly written in your paper that in your scaling/evaluation the traits are treated in other ways like in the linear scoring. You should underline that every change/deviation from correct position of limbs is evaluated as a separate trait. That is not in accordance with the common linear scoring when extremes in both directions are treated as the same trait.
It is very important for better understanding. It is very important and interesting breeding issue for all breeds. However, it should be also clearly stated that horses being “correct” are counted as the class/level in both traits (directions of evaluations).
Table 2 needs the description of traits in words, in the supplement file (not only in graphs in the main text). This is extremely important as in many countries it would be different traits/graphs, and underlining these differences and understanding them correctly is significant.
From the graphs – the differences between open hock – 0-2 are not visible enough from your graph. Closed hock is ok, so please check your graphs and prepare the supplement table with a description of traits 0-2 for every trait. Put special attention to the traits ”coon foot”-“broken and upright foot” and “sloping foot-club foot”.
The traits –walk and trot – being part of your paper are not described at all. Please give detailed information on these traits. How and when are they evaluated? What is taken into account by this evaluation? Who evaluated these traits? How many persons? What is the average and standard deviation of these traits? Describe this official evaluation in detail. They cannot be only mentioned in the table with results.
L 190-194 – please specify at least here what is it “animals selected for breeding and conformation”? at what age? What kind of “breeding”? The main horse register or parents of animals? Selected for confirmation seems not clear at all. The same selection procedures as you described in the table/graphs? Probably such separation or analysis should be in the methods section for better clearance and preparing the reader for your results.
The procedure GLZ should be better described in the method part as in table 4 you have “risk factors” – and it is not clear and mentioned earlier.
Please explain your methods – as you write in the text in the part on genetic correlations – standard deviation. L 216 and L 213-214 and L 398. The genetic correlations are usually evaluated with standard errors, not standard deviations.
Please reconsider the part on genetic correlations - it would be wise to write in the text on correlations about minus and plus values because minus values not always can be understood as negative in real meaning.
In the discussion part, another way of investigating traits in your and other papers on linear scoring should be underlined widely and strongly. It has a great meaning.
L 273 – trotting and walking connections should be discussed wider based on the information that has to be added to the material and methods part.
L 318 – this idea should be addressed strongly, especially in the context of different usage/dressage style of Menorca horses.
There is a lack of discussion part when the status of Menorca horses would be discussed. As far as I know, it is an endangered breed as the aim for the breed is to keep their genetic variability. So the heritability could be calculated, however with the other aim like in the open sport horse population. That should be underlined.
The sentence about the possibility of using breeding value estimation in conclusions because of high heritability should be excluded. That may be mentioned in the discussion that the heritability is meaningful, but using breeding value estimation and selection procedures based on EBV in endangered breeds is against its main endangered population aim – keeping genetic variability.
Your work is very useful in the part of genetic correlations and their meaning, but selection need in endangered population mean that it is not endangered anymore. So are Menorca horses endangered or not?
The limitation part is missing – the effect of the appraiser should be discussed there, subjective character of the data and small numbers of animals in calculations as well as a very small number of animals in pedigree. I even do not know if it is not a mistake. Only one generation back taken into account? Do you have 509 animals (L97) as evaluated for limb traits and 1017 animals in pedigree (L143)? It is even less than two parents for 509 animals.
Author Response
REVIEWER 3
The paper covers a very interesting subject on specific horse evaluation. The paper is well written and well structured. However, some important items have to be corrected and explained. The main issue is connected with traits presentation and lack of the limitation part in the discussion. Table 1/ (figure rather) with graphical traits description traits seems not clear in all cases, evaluation and conditions of walk and trot, being also evaluated traits, are not described. Some sentences need explanation.
Answer: Authors wish to thank you for your help to improve this manuscript. All the changes realized following the recommendations of the reviewed are included in the new version using red color. Another table has been included, as supplementary table S1, to clarify the description of the analyzed defects. And the description of the functional scores for walk and trot has also been included in the description of the material (lines 114-119).
In detail:
L 72-75 Please specify the movements and the practical source of their description, these terminology – bot, spine do not come from FEI dressage rules – and you call your horse - dressage horse.
This terminology –spine - can be seen as Western riding connected – as spine is a Western term, and the readers should know what you mean exactly. Your explanations even interesting, are too short and not enough. Be more clear in the paper and give citations to the Menorca horse dressage rules/or detailed descriptions of movements you describe.
Answer: More information about Menorcan Dressage has been included in the introduction section of the new version (lines 72-75). Term “spin” was used as a reference to the paper published by Sole et al (2013 - doi:10.1016/j.jevs.2012.12.002- and 2014 - doi:10.5424/sjar/2014121-4686-). It has been changed by “pirouette” to clarify the information (lines 77-78).
Table 1 – it should be clearly written in your paper that in your scaling/evaluation the traits are treated in other ways like in the linear scoring. You should underline that every change/deviation from correct position of limbs is evaluated as a separate trait. That is not in accordance with the common linear scoring when extremes in both directions are treated as the same trait.
It is very important for better understanding. It is very important and interesting breeding issue for all breeds. However, it should be also clearly stated that horses being “correct” are counted as the class/level in both traits (directions of evaluations).
Answer: More information about the used scale has been included in the methodology section of the new version to clarify the indicated issues (title of table 1 and lines 109-111).
Table 2 needs the description of traits in words, in the supplement file (not only in graphs in the main text). This is extremely important as in many countries it would be different traits/graphs, and underlining these differences and understanding them correctly is significant.
From the graphs – the differences between open hock – 0-2 are not visible enough from your graph. Closed hock is ok, so please check your graphs and prepare the supplement table with a description of traits 0-2 for every trait. Put special attention to the traits ”coon foot”-“broken and upright foot” and “sloping foot-club foot”.
Answer: Graphic representation of the open hock has been changed in table 1 of the new version. Written description of the analyzed traits is also included in table 2 and a supplementary table has been included with the written description of the limb conformation defects analyzed to clarify the information to the readers (see Supplementary table S1).
The traits –walk and trot – being part of your paper are not described at all. Please give detailed information on these traits. How and when are they evaluated? What is taken into account by this evaluation? Who evaluated these traits? How many persons? What is the average and standard deviation of these traits? Describe this official evaluation in detail. They cannot be only mentioned in the table with results.
Answer: More information about the evaluation of walk and trot traits is included in the methodology section of the new version (lines 114-119).
L 190-194 – please specify at least here what is it “animals selected for breeding and conformation”? at what age? What kind of “breeding”? The main horse register or parents of animals? Selected for confirmation seems not clear at all. The same selection procedures as you described in the table/graphs? Probably such separation or analysis should be in the methods section for better clearance and preparing the reader for your results.
Answer: This term is referred to the studs which select their animals for purchase and/or replacement using breeding and breed quality/conformation criteria. The age can change if the animals included in the official Studbook (main horse register) are selected for purchase (at different ages) or replacement (at birth).
Animals are selected for conformation using more complete information. From a conformational point of view, you can check the selected criteria used in Perdomo-González et al. (2022; https://doi.org/10.3390/2ani12182319). The information related with selection criteria is included in the material and methods section of the new version (lines 225-227) and also in the results section (lines 214-215).
The procedure GLZ should be better described in the method part as in table 4 you have “risk factors” – and it is not clear and mentioned earlier.
Answer: The procedure has been better described in methodology section (lines 133-139). In the title of table 4 “risk factors” has been changed by “non-genetic effects”, those that influence the analyzed defect traits.
Please explain your methods – as you write in the text in the part on genetic correlations – standard deviation. L 216 and L 213-214 and L 398. The genetic correlations are usually evaluated with standard errors, not standard deviations.
Answer: The standard error is used to quantify the precision of a point estimate in frequentist statistics, in Bayesian models the standard deviation is used to represent the uncertainty around parameter values as a probability distribution. That is the reason to include the standard deviations. You can obtain more information about the Bayesian models in the publication of van de Schoot et al. (2014; https://doi.org/10.1111/cdev.12169).
Please reconsider the part on genetic correlations - it would be wise to write in the text on correlations about minus and plus values because minus values not always can be understood as negative in real meaning.
Answer: The term “negative” related to the genetic correlations is replaced for “minus values” in the new version (lines 236, 241 and 429).
In the discussion part, another way of investigating traits in your and other papers on linear scoring should be underlined widely and strongly. It has a great meaning.
Answer: More information about different methodologies applied to investigate conformational defects in horses has been included (lines 283-294).
L 273 – trotting and walking connections should be discussed wider based on the information that has to be added to the material and methods part.
Answer: More information about walking and trotting have been included both in material and methods part (lines 114-119) and discussion section (lines295-301).
L 318 – this idea should be addressed strongly, especially in the context of different usage/dressage style of Menorca horses.
Answer: Authors agree with the importance of this information, nevertheless, currently there is not enough information about Menorcan Dressage’s scores for the animal with limb conformations defects. Only around the 14% of the animals included in this analysis have performance data available for dressage ability.
Future analysis may be realized in this sense to analyze the influence of limb conformation defects on the results obtained by the animals in the official competitions. This is the reason why only scores at walk and trot, obtained in the corresponding morphological events by the same appraisers, were used in the current analysis. Thank you very much for your suggestion.
There is a lack of discussion part when the status of Menorca horses would be discussed. As far as I know, it is an endangered breed as the aim for the breed is to keep their genetic variability. So the heritability could be calculated, however with the other aim like in the open sport horse population. That should be underlined.
The sentence about the possibility of using breeding value estimation in conclusions because of high heritability should be excluded. That may be mentioned in the discussion that the heritability is meaningful, but using breeding value estimation and selection procedures based on EBV in endangered breeds is against its main endangered population aim – keeping genetic variability.
Answer: The sentence about using breeding values has been deleted from conclusions. The discussion has been modified to include information about the Menorca horse status and its relationship with its breeding program. (Lines 453-461).
Your work is very useful in the part of genetic correlations and their meaning, but selection need in endangered population mean that it is not endangered anymore. So are Menorca horses endangered or not?
Answer: Yes, the Menorca horses are recognized as an endangered population and their official breeding program included measures for the selection and the conservation of the population. It is a mixed program, as you can see in the following link to the official web-site of the Spanish Ministry of Agriculture, Fisheries and Food: https://www.mapa.gob.es/es/ganaderia/temas/zootecnia/en_programa_de_cria_prme_definitivo_tcm30-561612.PDF_. Please, see page 12, where there is a clear indication about this issues, as follow:
General Objective: Attain horses with a conformation—in keeping with the established breed prototype—adapted to their functional aptitudes and with qualities that facilitate their handling for Dressage while at the same time allowing them to become outstanding in the disciplines for which they have been selected, maintain the existing genetic variability and favoring the general growth of herd numbers.
Specific Objectives: A) Attain horses with certain morphological and conformational traits that favor the indirect improvement of their functional performance (functional conformation), but always within the establish breed prototype. B) Attain horses with an adequate functional aptitude to excel in competitions for various disciplines such as Dressage and Menorcan Dressage, both at the national and international level. C) Select horses while maintaining genetic variability at all times, minimizing endogamy and kinship of the population to guarantee the preservation of the breed and increase general census numbers to avoid extinction.
The limitation part is missing – the effect of the appraiser should be discussed there, subjective character of the data and small numbers of animals in calculations as well as a very small number of animals in pedigree.
Answer: All the limitations have been included in the new version of the manuscript (lines 447-451). The appraisers have been trained before data collection and they are also periodically tested (lines 108-109). Nevertheless, we consider that the limited number of appraisers (3) can also contribute to common evaluation criteria and better data quality.
The reduced number of sampled animals can be justified by the endangered status of the population (lines 453-455). The official census of PRMe animals located in Spain at December 31, 2023 was 2863 individuals, and 1681 of them are between 3 and 20 years-old. Therefore, the 29.92% of them have been included in the analysis.
Finally, the small number of animals in the pedigree can be explained by the average maximum known generations (2.5, lines 167-168).
I even do not know if it is not a mistake. Only one generation back taken into account? Do you have 509 animals (L97) as evaluated for limb traits and 1017 animals in pedigree (L143)? It is even less than two parents for 503 animals.
Answer: The Menorca horses’ populations is small, with many sampled animals being siblings or having parent-offspring relationships, resulting in a limited number of animals in the pedigree file.
Thanks this, the population has shown significant heritability estimates for all analyzed traits, with low standard deviation values, and 64.30% of estimated genetic correlations having a standard deviation value lower than the genetic correlation itself.
The Menorca horses’ population is a healthy horse population in which genetic variability is monitored annually, using pedigree and molecular data to guide breeders in the genetic management. However, challenges exist due to the small size of the selection nucleus, limited replacement and long generational intervals. Implementation of selection procedures is important to increase the number of desirable stallions and mares, and to encourage early replacement of breeding stock by breeders, while maintaining genetic variability.
Round 2
Reviewer 3 Report
Comments and Suggestions for Authors
The paper was improved. All suggestions were taken into account by the Authors.